# Estimating the Influences of Prior Residual Stress on the Creep Rupture Mechanism for P92 Steel

**Dezheng Liu** [1,*] , **Yan Li** [1], **Xiangdong Xie** [2,*], **Guijie Liang** [1] **and Jing Zhao** [1]

1   Department of Mechanical Engineering, Hubei University of Arts and Science, Xiangyang 441053, China; wustliyan@163.com (Y.L.); guijie-liang@hotmail.com (G.L.); zjjysu@163.com (J.Z.)
2   School of Urban Construction, Yangtze University, Jingzhou 201800, China
*   Correspondence: liudezheng126@126.com (D.L.); xdxie@yangtzeu.edu.cn (X.X.);
    Tel.: +86-1869-620-3128 (D.L); +86-1560-861-9775 (X.X.)

**Abstract:** Creep damage is one of the main failure mechanisms of high Cr heat-resistant steel in power plants. Due to the complex changes of stress, strain, and damage at the tip of a creep crack with time, it is difficult to accurately evaluate the effects of residual stress on the creep rupture mechanism. In this study, two levels of residual stress were introduced in P92 high Cr alloy specimens using the local out-of-plane compression approach. The specimens were then subjected to thermal exposure at the temperature of 650 °C for accelerated creep tests. The chemical composition of P92 specimens was obtained using an FLS980-stm Edinburgh fluorescence spectrometer. Then, the constitutive coupling relation between the temperature and material intrinsic flow stress was established based on the Gibbs free energy principle. The effects of prior residual stress on the creep rupture mechanism were investigated by the finite element method (FEM) and experimental method. A comparison of the experimental and simulated results demonstrates that the effect of prior residual stress on the propagation of micro-cracks and the creep rupture time is significant. In sum, the transgranular fracture and the intergranular fracture can be observed in micrographs when the value of prior residual stress exceeds and is less than the material intrinsic flow stress, respectively.

**Keywords:** residual stress; creep rupture mechanism; P92 steel; FEM; Gibbs free energy principle

## 1. Introduction

Creep deformation and failure in high-temperature structures are serious problems in industry and are becoming even more serious under the current increasing pressures of energy, economics, and sustainability [1–3]. P92 high Cr alloy steels are widely used as high-temperature construction materials in power plants due to their high creep strength, good molding property to be processed, and superior heat properties [4,5]. However, such components employed in power plants are continually exposed to high temperatures and high steam pressures, and creep crack growth could occur within these high-temperature regimes, causing the failure of these components [6]. In practical engineering, it is difficult to accurately evaluate the creep rupture mechanism of P92 steels under multi-axial stress states due to the intricate variations of the stress, strain, and damage at the tip of a creep crack with time [7]. For example, residual stresses can be invariably introduced into structural components during the fabrication processes and thermal operations and non-uniform plastic deformation [8,9], and the residual stresses can be superimposed by any applied loading and produce a complex stress state acting on in-service components, causing creep crack growth and rupture. Therefore, it is essential to investigate the influences of residual stress on the creep rupture mechanism for high Cr alloy steel.

In recent years, the creep rupture behavior of high Cr alloy steel has been studied extensively. For example, recent studies [10–15] have reported the prediction of the creep damage and lifetime

for P92 steels based on the modification of existing creep damage models. However, there are few detailed studies of the effect of prior residual stress on the creep rupture mechanism of P92 steels. The difficulty in describing the creep rupture behavior of P92 steel is due to the lack of an accurate rupture mechanism for depicting the microscopic crack characteristics [16]. Creep fracture is usually caused by the growth of nucleation and mutual connection of micro-cavities and micro-cracks, and creep cracks grow from the cusp and ultimately weaken the cross section to the point where failure occurs [13]. Due to the fact that residual stresses can be generated during the fabrication processes for many mechanical or thermal operations, creep crack initiation and growth in components can be driven by these residual stresses [8]. Especially under multi-axial stress states, the growth of the crack-like defects, such as micro-cracks and creep voids, can be accelerated by the residual stresses combined with external applied loading. Furthermore, the residual stress can contribute to the amplitudes of crack tip stress fields and the cracks will be nucleated when the accumulated creep strains under the action of the stress field around the crack tip are sufficient to exhaust the creep ductility of the material [9]. Therefore, the residual stress may significantly affect the load carrying capacity and resistance to creep and rupture of the structural components made of P92 steel.

To better understand the creep rupture mechanism of residual stress and initial crack positions and to accurately analyze creep failure and life assessments, it is essential to quantitatively investigate the effect of prior residual stress levels on creep crack growth and rupture. There are two main ways to introduce residual stresses into measured specimens. One way is to extract a specimen from the structure containing residual stresses and then introduce a flaw in the region of the residual stress, and the other one is to directly introduce residual stresses into specimens through the application of either mechanical or thermal techniques [17]. The local out-of-plane compression approach [17], which is the use of mechanical techniques and the most well-known residual stress generation method, is adopted in this study. Furthermore, the numerical methods based on the damage model combined with experimental tests can provide an efficient way to analyse the effect of stress levels on creep crack growth and rupture. It should be noted that some researchers [18,19] have attempted to describe the process of high-temperature deformation of metallic materials through the use of material intrinsic flow stress. In reference [18], the thermal deformation behavior and the hyperbolic constitutive equation of T122 steel were investigated by the determination of deformed activation energy and material intrinsic flow stress. In reference [19], the relationship between residual stress and material intrinsic flow stress has been used to determine the formation of weld solidification cracks. On the basis of references [18,19], the method for exploring the effects of prior residual stress on the creep rupture mechanism through the comparison of residual stress and material intrinsic flow stress was conducted in this study.

This paper is organized as follows. Reviews of previous literature on the investigation of the relationship between the residual stress and creep rupture behavior for P92 steel were presented previously. In Section 2, the methodologies of creep crack growth test are conducted, involving methods of introducing different residual stresses for creep tests and establishing the constitutive coupling relation between the temperature and material intrinsic flow stress, followed by the construction of the corresponding finite element (FE) model. The results and discussions of the influences of prior residual stress on the creep rupture mechanism for P92 steel are presented in Sections 3 and 4, respectively. Conclusions are drawn in Section 5.

## 2. Materials and Methods

### 2.1. Specimen Preparation and Creep Test

The specimens of P92 steel were provided by Angang Steel Company (Anshan, China) Limited. In order to introduce the multi-axial stress states into the specimens for the creep test, the annular breach in the middle of each specimen was machined by the lathe. The geometry of the notched specimen is shown in Figure 1. The width at both ends of the specimen is 30 mm, and the width at the

middle of the specimen is 20 mm. The annular breach radius is set to 0.7 mm and the thickness of the specimen is 2 mm. The notch sharpness is set to 28.6, which is the ratio of the width of the middle segment to the radius of the notch. The chemical composition of the P92 specimen was measured using an FLS980-stm Edinburgh fluorescence spectrometer (Edinburgh Instruments Ltd, Livingston, UK) and the chemical composition is presented in Table 1. To analyze the evolution of the creep fracture surface, there are five samples under different residual stress levels, respectively.

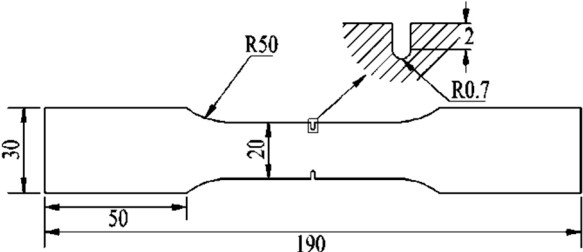

**Figure 1.** Schematic diagram of the specimen (mm).

**Table 1.** Chemical compositions of P92 steel (wt.%).

| Material | C | Mn | Si | Cr | Mo | S | P | Nb | V | Al | Ni | W | N | B | Bal. |
|---|---|---|---|---|---|---|---|---|---|---|---|---|---|---|---|
| P92 steel | 0.1 | 0.45 | 0.35 | 8.95 | 0.96 | 0.01 | 0.018 | 0.08 | 0.215 | 0.04 | 0.12 | 1.47 | 0.043 | 0.001 | Fe |

The Gibbs free energy interface thermodynamic method [20] was adopted to establish the constitutive coupling relation between the temperature and material intrinsic flow stress. The basic equation for the Gibbs energy of a multi-component solution phase can be expressed as

$$G_m = \sum_i x_i G_i^0 + RT \sum_i x_i ln x_i + \sum_i \sum_{j>i} x_i x_j \sum_V \Omega_V \left( x_i - x_j \right)^V \tag{1}$$

where $\sum_i x_i G_i^0$ is the Gibbs energy of the pure components, $RT \sum_i x_i ln x_i$ is the ideal entropy, and $\sum_i \sum_{j>i} x_i x_j \sum_V \Omega_V \left( x_i - x_j \right)^V$ accounts for pairwise interactions of the species. According to the Gibbs free energy principle [20], the equilibrium pressure of the system is achieved when the Gibbs's free energy attains a minimum. The solute mass fraction in the solid [21] can be expressed as

$$\tau\left( X, T \right) = \frac{1}{\beta \, 2^{\frac{N}{8}} D \, \Delta \, T^q} \int_0^x \frac{dx}{x^{\frac{2(1-x)}{3}} \cdot \left( 1 - x \right)^{\frac{2x}{3}}} \tag{2}$$

where $\tau$ is the solute mass fraction in the solid for different amounts of undercooling, $\beta$ is an empirical coefficient for describing the kinetics of isothermal austenite to pearlite reaction, $N$ is the size of a grain, $D$ is an effective diffusion coefficient, $\Delta T$ is the undercooling, $q$ is an exponent that depends on the effective diffusion mechanism, and $x$ is the fraction transformed.

In a high-temperature environment, the various microstructural constituents can significantly affect the mechanical properties and the distribution of residual stresses in components. Therefore, it is important to be able to predict the final microstructure distribution for a given thermal history. Equations (1) and (2) can be used to describe the characterization of the kinetics of austenite decomposition to equilibrium or non-equilibrium phases in alloy steel. The kinetics of isothermal austenite to ferrite, pearlite, and bainite reactions can also be described through the use of the Gibbs free energy interface thermodynamic method. However, the various phase transformations that generate the required microstructure link the process parameters to the final properties. For example, the empirical coefficient in Equation (2) is only to be used in the kinetics of isothermal austenite to pearlite reaction [21].

According to reference [20], the relationship between the stress and strain can be determined by the yield strength, hardening exponent, and reference strain, as follows:

$$\sigma = \frac{\sigma_{0.2}}{\varepsilon_0^n} \cdot \varepsilon^n \qquad (3)$$

where $\sigma_{0.2}$ is the yield strength, $n$ is the hardening exponent, and $\varepsilon_0$ is a reference strain. The determination of the value of the parameters in Equations (1)–(3) has been reported in references [21,22]. In this study, the value of the parameters used to derive the relationship between the temperature and the material intrinsic flow stress is shown in Table 2.

**Table 2.** The value of the parameters used to derive the material intrinsic flow stress.

| Parameter | $\beta$ | $N$ (um) | $D$ (m²/s) | $\Delta T$ (°C) | $q$ | $\sigma_{0.2}$ (MPa) | $n$ | $\varepsilon_0$ |
|-----------|---------|----------|------------|-----------------|-----|----------------------|-----|-----------------|
| Value | 0.3054 | 20 | $2.09 \times 10^{-12}$ | 123.1 | 3 | 440 | 0.129 | 1 |

Using Equations (1)–(3) and Tables 1 and 2, a constitutive relation between the temperature and material intrinsic flow stress can be obtained using JmatPro software (JmatPro 7.0, Sente Software Ltd, Surrey, UK).

Figure 2 shows the material intrinsic flow stress of P92 steel in the temperatures range from 0 °C to 1000 °C. According to Figure 2, the material intrinsic flow stress is about 145 MPa when the temperature is 650 °C. The residual stresses introduced into the specimens were devised as two levels; here, the value of material intrinsic flow stress is taken as the reference standard. One level of residual stress is above the material intrinsic flow stress, while the other one is below the material intrinsic flow stress.

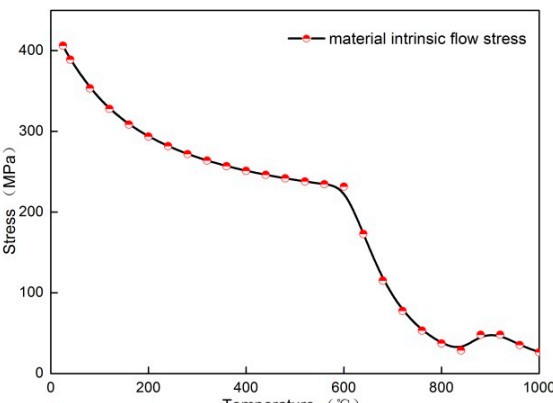

**Figure 2.** Material intrinsic flow stress vs. temperature.

The local out-of-plane compression approach [17] was used to introduce residual stress states in specimens. According to reference [17], the tensile residual stresses ahead of the annular breach can be generated by loading in compression beyond the yield and then unloading. Furthermore, the tensile residual stress can be changed through altering loading point displacement. The specimen surface is compressed until a specified displacement to produce some plastic deformation around the annular breach. In this study, the specimens with dimensions of 190 mm × 30 mm × 2 mm were used to introduce residual stress states through the use of the SUNS WAW-2000 universal material hydraulic experiment machine (Shenzhen Suns Technology Stock Co., Ltd, Shenzhen, China). The notched specimen was loaded in compression to introduce a tensile residual stress field over a significant distance ahead of the annular breach. Pre-strained specimens were produced by uniaxially straining large tensile specimens to −3.82, +2.15, −1.78, and +1.03 pct when the different plastic deformation levels ahead of the annular breach were produced. The specimen was then unloaded by moving

the tools back to their original position, and a residual stress field was generated due to the strain incompatibility between the elastic and plastic regions ahead of the annular breach.. In conjunction with the FERMI customized X-ray diffractometer thermal stage, the value of residual stress can be measured at the temperature of 650 °C through the use of a BRUKER D8 advance X ray diffractometer (Bruker Corporation, Ettlingen, Germany), as shown in Figure 3. Two levels of residual stress were introduced into the notched P92 specimens through the different loading point displacements.

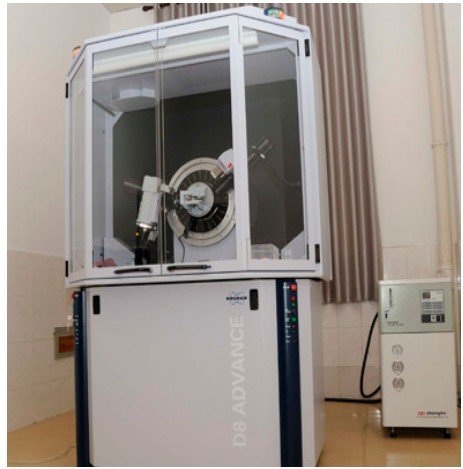

**Figure 3.** BRUKER D8 advance X ray diffractometer.

The parameters of the pre-compression to introduce the residual stress field into the P92 specimens are shown in Table 3. After the compression experiment of the specimens, creep tests for two groups of specimens with different residual stress levels were conducted at the temperature of 650 °C. In order to obtain creep crack initiation and propagation properties of specimens by an accelerated creep test, an external pulling stress of 110 MPa was applied to the specimens. The RDL50 electronic high-temperature creep and lasting intension testing machine (JiLin Guanteng Automation Technology Co., Ltd, Changchun, China) was adopted in the accelerated creep test, as shown in Figure 4.

**Table 3.** Parameters of the pre-compression to introduce the residual stress field.

| Residual Stress Level | Pre-Compression Load (KN) | Loading Speed (mm/min) | Compressing Magnitude (%) | Stretching Magnitude (%) | Maximum Residual Stress Value (MPa) | Material Intrinsic Flow Stress Value (MPa) |
|---|---|---|---|---|---|---|
| High level | 42.8 | 0.16 | 3.82 | 2.15 | 182 | 145 |
| Low level | 20.6 | 0.12 | 1.78 | 1.03 | 97 | 145 |

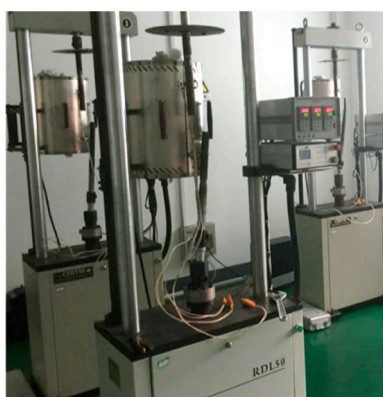

**Figure 4.** Accelerated creep tests for P92 specimens.

In accelerated creep tests, the temperatures were kept within ±1 °C during the test through the use of a thermocouple to measure the center of the specimen and the position of the upper and lower pull rods. Meanwhile, the displacement of loading line and the length of the crack were measured and recorded. The displacement of the loading line was measured by an extensometer with a measurement accuracy of 1 micron and the creep crack length was measured by the direct current potential drop method [17]. After the creep test, the specimen was dissected into two parts along with its axial direction of thickness by the wire-electrode cutting method. One part was used to observe the morphology of the crack growth surface, and the other one was used to observe the change of the microstructures.

## 2.2. Creep Damage and Crack Growth Model

The continuum damage mechanics (CDM) based creep damage model has been widely used to describe all three stages of creep because it can be easily implemented with the finite element program in the analysis of creep damage behavior, which can significantly improve the efficiency in terms of both time and economy. In applications, a damage parameter is defined through the use of the CDM-based approach that ranges from zero (no damage) to a critical damage value (full damage), and is then measured throughout the creep processes. Creep failure time is defined as the time taken for the continuum damage level to move from no damage to full damage [23]. Physically-based CDM for creep provides a suitable framework for quantifying the shapes of the creep curve caused by several microstructural damage mechanisms.

The computational capability for creep damage analysis relies on the availability of a set of creep damage constitutive equations. In this study, Hayhurst creep damage constitutive equations [24], which were developed based on the CDM approach, were used to investigate the influences of residual stress on the creep rupture mechanism for P92 steel. Hayhurst creep damage constitutive equations are shown in the following:

$$\dot{\varepsilon}_{ij} = \frac{3s_{ij}}{2\sigma_e} Asinh\left[\frac{B\sigma_e(1-H)}{(1-\varnothing)(1-\omega)}\right] \tag{4}$$

$$\dot{H} = \left(\frac{h\dot{\varepsilon}_e}{\sigma_e}\right)\left(1 - \left(\frac{H}{H^*}\right)\right) \tag{5}$$

$$\dot{\varnothing} = \left(\frac{K_c}{3}\right)(1-\varnothing)^4 \tag{6}$$

$$\dot{\omega} = CN\dot{\varepsilon}_e(\sigma_1/\sigma_e)^v \tag{7}$$

where $\dot{\varepsilon}_{ij}$ is the creep strain rate under the multi-axial stress state, $\varepsilon_{ij}$ is the multi-axial creep rate, $\sigma_1$ is the maximum principal stress, $\omega$ is the creep damage, $\dot{\omega}$ is the creep damage rate, $H$ is the strain hardening, $\dot{H}$ is the strain hardening rate, and $n$ represents the stress exponents. $N = 1$ when $\sigma_1 > 0$ and $N = 0$ when $\sigma_1 < 0$. The multi-axial parameters $A$, $B$, $C$, $h$, $H^*$ and $k_c$ are constants to be determined from the uniaxial creep behaviour [25]. Variable $\varnothing$ increases from its initial value of zero towards a theoretical upper limit of unity. The parameter $v$ is the multi-axial stress sensitivity index, where $\sigma_e = \left(3S_{ij}S_{ij}/2\right)^{1/2}$ is the effective stress, $S_{ij} = \sigma_{ij} - \delta_{ij}\sigma_{kk}/3$ is the stress deviator, and $\varepsilon_e = \left(2\varepsilon_{ij}\varepsilon_{ij}/3\right)^{1/2}$ is the effective creep strain. The creep damage parameter $\omega$ is defined ranging from 0% (no damage) to 100% (full damage) and the parameter is then monitored throughout the creep time. The creep rupture time is defined as the time taken for the continuum damage level to reach 99.9–100% in most elements in a given zone [24].

The behavior of creep crack growth can be evaluated in terms of the range of stress intensity factor, and the formula for describing the behavior of creep crack growth [26] can be written as

$$\Delta K = \frac{\Delta P}{B^{1/2}W^{1/2}}F(a/W) \tag{8}$$

$$\Delta P = (1-R)P_{max} \tag{9}$$

$$F(a/W) = \left[ \frac{2+a/W}{(1-a/W)^{3/2}} \right] \left( 0.886 + 4.64(a/W) - 13.32(a/W)^2 + 14.72(a/W)^3 - 5.6(a/W)^4 \right) \tag{10}$$

where $\Delta K$ is the stress intensity factor; $\Delta P$ is the range of applied loads; $F(a/W)$ represents the behavior of creep crack growth; $R$ is the ratio of maximum stress and normal stress; $a$ is the length of the crack; and $W$ and $B$ are the width and thickness of the specimen, respectively.

The fracture parameter of creep [26] can be written as

$$(C_t)_{avg} = \frac{\Delta P \Delta V_c}{B^{1/2} W t_h} \frac{F\prime}{F} \tag{11}$$

$$\frac{F\prime}{F} = \left[ \left( \frac{1}{2+a/W} \right) + \left( \frac{3}{2(1-a/W)} \right) \right] + \left[ \frac{4.64 - 26.64(a/W) + 44.16(a/W)^2 - 22.4(a/W)^3}{0.886 + 4.64(a/W) - 13.32(a/W)^2 + 14.72(a/W)^3 - 5.6(a/W)^4} \right] \tag{12}$$

where $(C_t)_{avg}$ is the fracture parameter, $\Delta V_c$ is the load line displacement caused by creep deformation, and $t_h$ is the loading time. In the process of creep crack growth, the total load line displacement is mainly composed of the displacement caused by creep deformation and the displacement caused by elastic deformation. Therefore, the displacement caused by creep deformation can be written as

$$\Delta V_c = \Delta V - \Delta V_e \tag{13}$$

$$\Delta V_e = \frac{t_h (da/dt)_{avg}}{P} B \frac{2(\Delta K)^2}{E\prime} \tag{14}$$

$$(da/dt)_{avg} = (1/t_h) \cdot (da/dN) \tag{15}$$

where $\Delta V$ is the total load line displacement, $\Delta V_e$ is the load line displacement caused by elastic deformation, $(da/dt)_{avg}$ is the average crack growth rate, and $E'$ is the effective modulus of elasticity. $E' = E/(1-v^2)$ under the plane strain state and $E' = E$ under the plane stress state. According to reference [27], the fracture behavior can be defined as brittle fracture when $\Delta V_e > \Delta V_c$, and the fracture behavior can be defined as ductile fracture when $\Delta V_e < \Delta V_c$.

The above mentioned equations have been used to simulate the creep damage behaviors of P92 steel under a multi-axial stress state using ABAQUS finite element software (ABAQUS 6.10, Dassault Systemes Simulia Corporation, Johnston, RI, USA). A user programmable function UMAT was written by FORTRAN to calculate the values of stress, strain, and damage in notched specimens. According to reference [9], the elasticity modulus and Poisson's ratio of P92 steel at 650 °C are 125 GPa and 0.3, respectively. According to references [25,28], the creep parameters of P92 steel at the temperature of 650 °C are shown in Table 4.

**Table 4.** Creep parameters of P92 steel at the temperature of 650 °C.

| Material Parameter | $A$ (MPa/h) | $B$ (MPa$^{-1}$) | $C$ | $H$ (MPa) | $H^*$ | $K_c$ (MPa$^{-3}$/h) | $E$ (GPa) | $v$ |
|---|---|---|---|---|---|---|---|---|
| Value | $2.21618 \times 10^{-9}$ | $3.473 \times 10^{-3}$ | $9.85 \times 10^{-2}$ | $2.43 \times 10^6$ | 0.5929 | $9.227 \times 10^{-4}$ | 125 | 0.3 |

### 2.3. FE Analysis Model

The FE analysis model was built using the HyperWorks software (HyperWorks 11.0, Altair Corporation, Troy, MI, USA) for simulating the creep rupture process for the P92 notched specimen. The geometry and dimensions of the notched tension specimen were taken to be the same as those of the experiments, as shown in Figure 1. The simulated temperature was set to 650 °C, and the elasticity modulus and Poisson's ratio of the FE model were set to 125 GPa and 0.3, respectively. The FE mesh is shown in Figure 5, and an external tensile stress of 110 MPa was applied to the specimen.

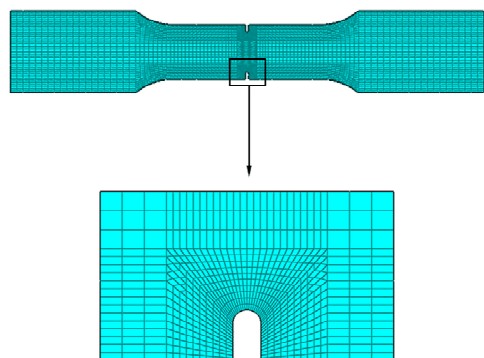

**Figure 5.** FE model of the notched P92 specimen.

The brick mesh (type C3D8R) and prismatic mesh (type C3D6) were used to model the evolution of creep rupture. The global size of the mesh was 1 mm, and the FE mesh was refined in the vicinity of the notch tip to obtain accurate results and to eliminate mesh dependency effects in the analyses. According to the average grain size of P92 steel [29], the smallest mesh size was set to 20 um. In this FE model, the number of elements and nodes were 18,677 and 24,136, respectively. Subsequently, the FEA model was imported into the ABAQUS software. The model was loaded by a uniform tensile stress at one end of the specimen and was fixed by an encastre constraint at the other end of the specimen, and the accumulated creep damage variables such as stress, strain, and damage could be calculated in ABAQUS using the corresponding user defined subroutine Creep UMAT. A flowchart used to link the different software between Hyperworks and ABAQUS is shown in Figure 6.

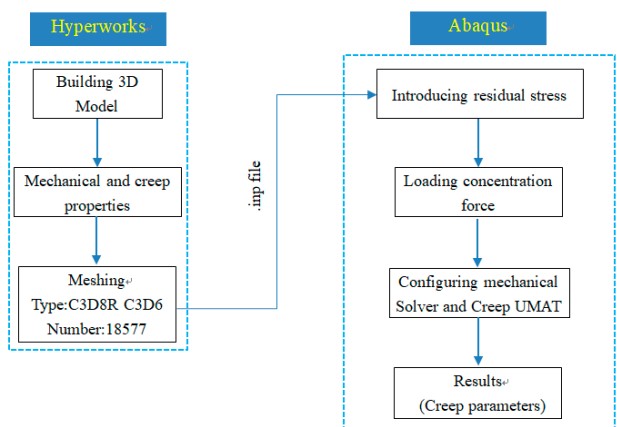

**Figure 6.** The flowchart used to link Hyperworks and ABAQUS.

For creep damage problems in FEM, the resulting equations are highly non-linear and stiff in nature. The nature of creep damage analysis is time dependant and the field variables such as stress, strain, and creep damage need to be updated where an integration scheme needs to be implemented. The stability and accuracy of the FE solution critically depends on the selection of the time step size associated with an appropriate integration method [30]. In this study, the well-known Runge-Kutta-Merson method [30] was adopted to control the time step because it can minimize the requirement of extra storage and reduce the amount of round-off error in creep damage analysis.

The FE analysis procedure was carried out in four stages: (1) the pre-compression was applied on the notched P92 specimens so as to produce the initial residual stress field; (2) a creep deformation modeling step at 650 °C was applied directly in order to predict relaxation, creep damage, and crack initiation under the different residual stress fields; (3) a primary load was applied in the FE models to investigate the combined effects of the residual stress with a primary load on the creep damage and crack initiation; (4) a creep loading step was applied and the integration time step size was controlled by the Runge-Kutta-Merson method.

## 3. Results

The creep strain and displacement of the whole rupture evolution of the P92 specimen from a constant pressure (110 MPa) test at a constant temperature of 650 °C were simulated through the implantation of Equations (4)–(15) in ABAQUS software by using the user subroutine UMAT. Figures 7 and 8 show the evolution of creep strain with the high and low residual stress level, respectively. When a specimen is stressed, its lattice spacing is altered. In the process of creep crack growth, the total load line displacement is mainly composed of the displacement caused by creep deformation and elastic deformation. The elastic strain in the material is determined by the change in crystal lattice spacing using Bragg's Law [31] and the load line displacement caused by creep deformation can be calculated by ABAQUS software.

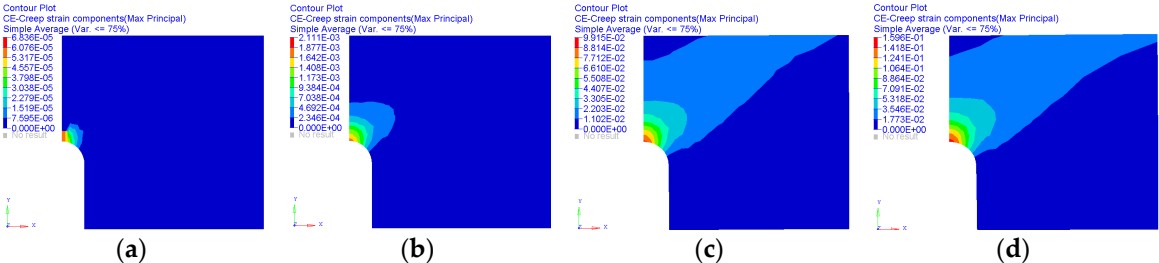

**Figure 7.** The evolution of creep strain at different lifetime fractions of the specimen with the high residual stress level. (**a**) 0%, (**b**) 50%, (**c**) 80%, and (**d**) 100%.

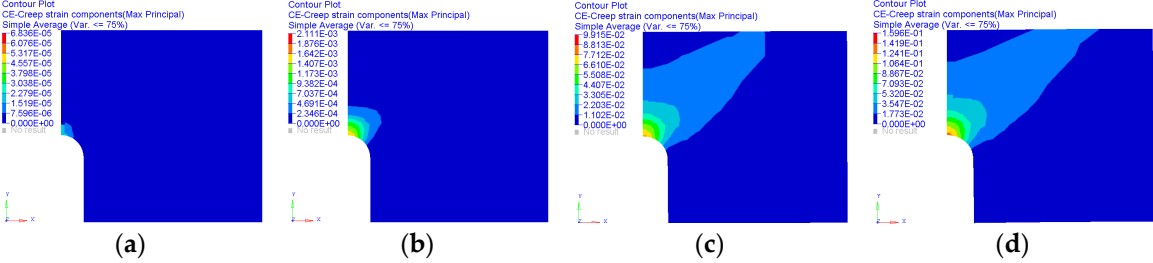

**Figure 8.** The evolution of creep strain at different lifetime fractions of the specimen with the low residual stress level. (**a**) 0%, (**b**) 50%, (**c**) 80%, and (**d**) 100%.

The effect of residual stress on creep deformation can be examined by comparing the predicted creep strain of the specimen with the introduced prior residual stress level higher than the material intrinsic flow stress level to that of the specimen containing the prior residual stress level less than the material intrinsic flow stress level. Figures 7 and 8 show the predicted evolution in equivalent creep strain with creep time from a constant pressure (110 MPa) test at a constant temperature of 650 °C under two different pre-residual stress levels. According to Figures 7 and 8, the strain ahead of the notch is increasing with the increase of creep time, and the maximum strain always appears near the notch tip. By extracting the creep displacement from the FE simulated results, the elastic displacement can be computed by subtracting the creep displacement from the total load line displacement measured. The evolution of load line displacement under different residual stress levels can be drawn in Figure 9.

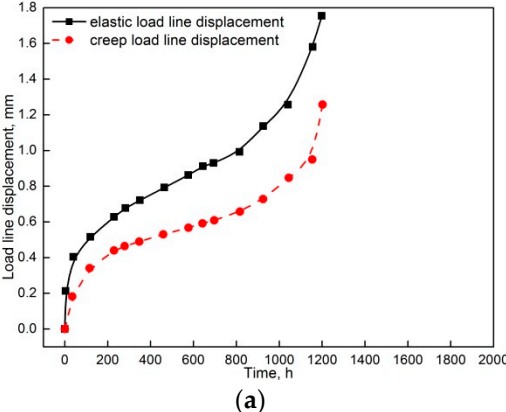 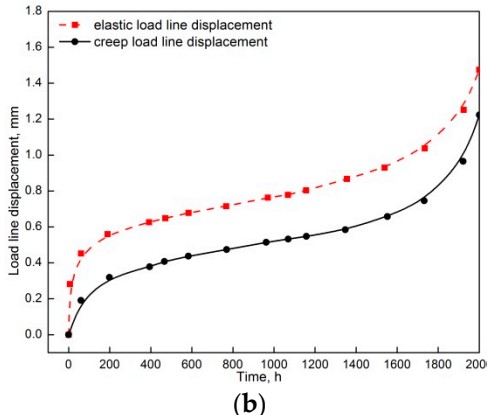

(a)    (b)

**Figure 9.** The evolution of the load line displacement caused by elastic deformation and creep deformation under different residual stress levels. (**a**) High residual stress level and (**b**) low residual stress level.

Figure 9 shows the evolution of the simulated load line displacement caused by elastic deformation and creep deformation under the different residual stress levels. It can be seen from Figure 9a that the simulated load line displacement caused by creep deformation is obviously higher than that of elastic deformation under the introduced residual stress level higher than the material intrinsic flow stress level. According to Figure 9b, the simulated load line displacement caused by creep deformation is obviously smaller than that of elastic deformation under the introduced residual stress value level less than the material intrinsic flow stress level.

To investigate the effects of the residual stress on creep rupture behavior, the evolution of the creep fracture surface under the different residual stress levels was studied. Figure 10 shows the microscopic morphology of creep crack distribution at the lifetime fraction of 90% under two different prior residual stress levels. There is an obvious difference among the overall morphologies of crack distribution at different prior residual stress levels. The rough and zig-zag growth path of the crack propagation can be observed in Figure 10a. By contrast, the relative smooth and straight growth path of the crack propagation can be observed in Figure 10b.

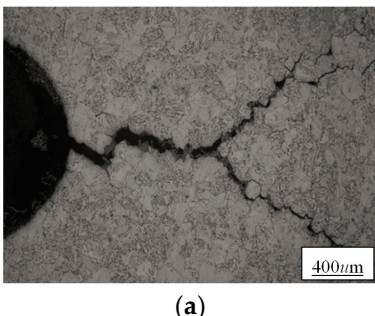 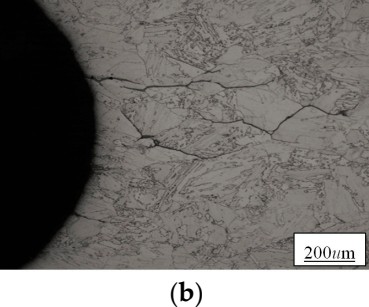

(a)    (b)

**Figure 10.** Creep crack distribution at the lifetime fraction of 90%. (**a**) The introduced prior residual stress level higher than the material intrinsic flow stress level and (**b**) the introduced residual stress value level less than the material intrinsic flow stress level.

Figure 11 shows the SEM images of the creep fracture surface near the annular breach of the samples along with the axial direction of thickness with different residual stress levels. There is an obvious difference among the overall morphologies at different lifetime fractions of the specimen. As shown in Figure 11a, the specimen subjected to pure local out-of-plane compression loading shows a smooth and cleavage fracture surface. With an increase of time, the microstructure of the fracture surface has obviously changed when the specimen is subjected to a constant tensile stress of 110 MPa at the temperature of 650 °C. It can be seen from Figure 11b that the fracture surface tends to be rough

and some small voids were formed on the fracture surface. As shown in Figure 11c,d, dimples are visible on the fracture surface. Figure 11c shows that a high tongue-shaped cleavage fracture occurred on the fracture surface, a ligament with a width between 5 um and 10 um appeared around the tip of the micro-crack, and a quasi-cleavage surface with obvious plastic deformation was observed. According to Figure 11d, the width of the ligament was increased up to about 20 um and some obvious cleavage steps occurred on the cleavage fracture sector. Moreover, the wedge shape cavities can be clearly observed, cavities nucleate at inclusions, and the combination of time and plasticity promotes their growth.

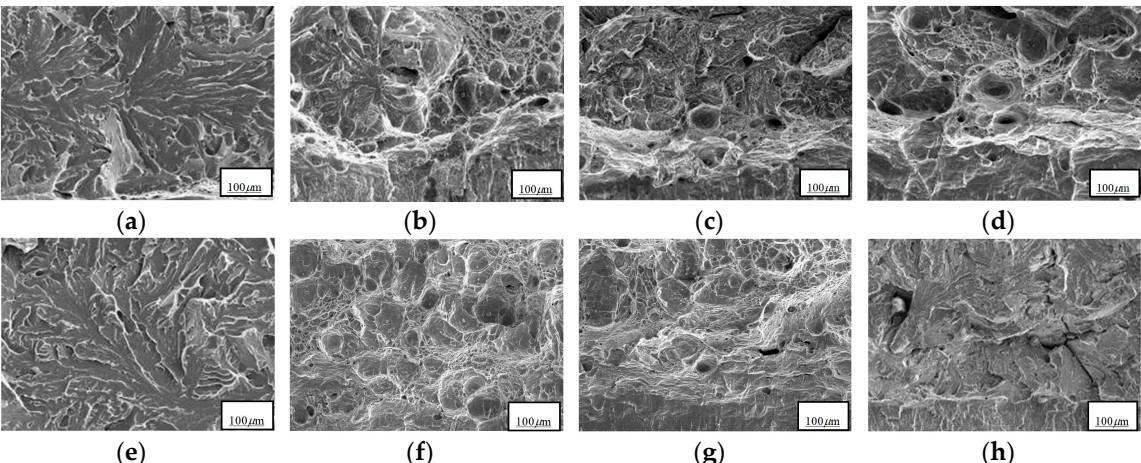

**Figure 11.** Creep fracture surface at different lifetime fractions of the specimen with different residual stress levels: high residual stress level (**a–d**): (**a**) 0%, (**b**) 50%, (**c**) 80%, and (**d**) 100%; and low residual stress level (**e–h**): (**e**) 0%, (**f**) 50%, (**g**) 80%, and (**h**) 100%.

As shown in Figure 11e–h, an obvious difference among the overall morphologies at different lifetime fractions of the specimen with the introduced prior residual stress value level less than the material intrinsic flow stress level was also observed. It can be seen from Figure 11e that dimples occurred on the quasi-cleavage surface. The evolution of lath boundaries from a low-angle to high-angle is apparent when the specimen is subjected to the creep loading, as shown in Figure 11f. At the lifetime fraction of 50%, voids can be obviously observed and the size of these voids is apparently smaller than that of the value of the prior residual stress that exceeds the material intrinsic flow stress through a comparison of Figure 11c,g. According to Figure 11f–h, the spheroidal-shape creep voids can be observed, the growth of the void by creep becomes slow, and many small voids are nucleated as a result of the fracture. Furthermore, the coalescence of discrete voids on prior grain boundaries were oriented approximately to the tensile stress axis. The width of the ligament was about 30 um and non-plastic lacerations were observed from Figure 11h.

## 4. Discussion

Using the FE analysis model described in Section 2.3, the creep strain and displacement of the whole rupture history of the P92 specimen under the two different residual stress levels were calculated and the predicted creep strain evolutions (Figures 7 and 8) indicate that larger creep strain accumulated in the vicinity of the notch tip during the stress relaxation process. However, the difference in the maximum creep strain value between the two different residual stress levels is obvious. With redistribution of the stress and strain during crack growth, the higher stress triaxiality could accelerate strain accumulation because the creep deformation was constrained by the hard crack growth. This means that residual stress could significantly affect the creep properties. According to the formula for describing the behavior of creep crack growth [27,28] in Section 2.2, the simulated load line displacement caused by elastic deformation and creep deformation under the different residual

stress levels (Figure 9) indicated that the ductile fracture is predominant when the value of the prior residual stress exceeds the material intrinsic flow stress, while the brittle fracture is predominant when the value of the prior residual stress is less than the material intrinsic flow stress.

The cavitation process, which includes creep voids' nucleation, growth, and linkage, is the most important factor that weakens the material and results in rupture. Using the experimental method described in Section 2.1, the evolution of the creep fracture surface under the different residual stress levels was investigated. SEM images of the creep fracture surface in Figure 11 indicate that the creep rupture process under the two different residual stress levels was obviously different. Figure 11a–d shows the process of void growth associated with a wedge-type cavity at the triple point of the grain boundary, which indicates the degradation of creep ductility [32]. According to references [32,33], evolution of the creep fracture surface under the high residual stress level in Figure 11a–d proves that transgranular cracking has resulted in ductile failure and the plasticity-controlled cavity growth is the predominant mechanism. Figure 11c,f–h shows that spheroidal-shape creep voids can be observed and the growth of the void by creep occurs slowly. Many small voids are nucleated as a result of fracture, and the small spheroidal-shape creep voids can be effectively explained by the brittle mechanism [34]. According to references [34,35], evolution of the creep fracture surface under the low residual stress level in Figure 11e–h proves that intergranular dimple mixed cracking has resulted in brittle failure and nucleation-controlled constrained cavity growth is the predominant mechanism.

Through the FE simulation of the evolution of creep deformation variables, the load line displacement caused by creep deformation elastic deformation can be obtained. According to the formula employed for describing the behavior of creep crack growth [27,28], the fracture behavior can be defined as brittle and ductile fracture when the elastic load line displacement is larger and less than the creep load line displacement, respectively. Thus, the failure type of the P92 specimen under different prior residual stress levels can be determined by comparing the simulated load line displacement caused by elastic deformation and the simulated load line displacement caused creep deformation. Furthermore, the SEM images of the creep fracture surface show that failure types were observed as ductile failure when the introduced residual stress level was higher than the material intrinsic flow stress level and brittle failure when the introduced residual stress value level was less than the material intrinsic flow stress level. It can be observed that the FE simulated results agree well with experimental results. Based on the Gibbs free energy principle [20], the constitutive coupling relation between the temperature and material intrinsic flow stress can be established. Unlike the modification of current creep damage models based on creep test results to predict the life of high-temperature components in previous studies [10–15], the main innovative method in this study is the estimation of the influences of residual stress on the creep rupture mechanism through a comparison of stress levels between the prior residual stress and the material intrinsic flow stress. FE simulations and experimental observations have proven that the new method for assessing the creep rupture mechanism under different prior residual stress levels presented in this study is reliable.

## 5. Conclusions

In summary, this study has investigated the effect of prior residual stress on the creep rupture mechanism for P92 steel. The different prior residual stress levels were introduced into the P92 specimens, were are then subjected to thermal exposure at the temperature of 650 °C for accelerated creep tests. At the same time, the constitutive coupling relation between the temperature and material intrinsic flow stress was established based on the Gibbs free energy principle through the extraction of the chemical composition of the P92 specimen. The evolution of the creep rupture process under the different prior residual stress levels was studied by FEM and experimental observations. FE investigations for determining the creep failure type in the P92 specimen under different prior residual stress levels were conducted through the use of the existing formula for describing the behavior of creep crack growth, and a comparison of the creep failure mechanism between the FE simulations and experimental results was then conducted. The FE simulated results agree well with

experimental results. It was found that when the introduced prior residual stress level is higher than the material intrinsic flow stress level, the transgranular cracking results in ductile failure and the plasticity-controlled cavity growth is the predominant mechanism. Additionally, the intergranular dimple mixed cracking results in brittle failure and nucleation-controlled constrained cavity growth is the predominant mechanism when the introduced prior residual stress level is less than the material intrinsic flow stress level. Moreover, it also demonstrates that material intrinsic flow stress can be used to assess the creep rupture mechanism under the different prior residual stress levels.

**Author Contributions:** Conceptualization, D.L.; methodology, D.L. and J.Z.; software, D.L. and Y.L.; validation, Y.L. and G.L.; formal analysis, X.X. and Y.L.; writing, D.L.; supervision, X.X.

**Funding:** This research was funded by the Hubei Superior and Distinctive Discipline Group of "Mechatronics and Automobiles" (No. XKQ2019009) and Hubei Key Laboratory of Power System Design and Test for Electrical Vehicle.

**Conflicts of Interest:** The authors declare no conflict of interest.

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
