# Peer review of "Estimating the Influences of Prior Residual Stress on the Creep Rupture Mechanism for P92 Steel"

_metals, doi:10.3390/met9060639_

Round 1

Reviewer 1 Report

line 102 108: improve the typing quality of the formulas. It is not possible to read them.

line 126: please, insert a scheme of the out-of-plane compression experimental arrangement used for the pre-stress and creep tests.

Please, clarify if  the measurement of the residual stresses have been carried out at room temperature.

Figure 9: The fracture images are taken from the half of the compression-crept specimens? Please, specify where the fracture is located , if there is only one  fracture surface and add some detail about the location of the  fracture images reported with respect to the totale fracture surfaces.

in Figure 9a, you speak about a dimpled fracture, but in it is possible to detect mainly clivage fracture, am I wrong? 

Please, add a compression creep experimental scheme in order to understand where the cracks are located.

How many specimens per condition? There is a experimental statistic?

line 365-375: add some more comment about the experimental and simulated results. It is not clear to me the FEM contribution to the research results and modellization.

General comment: Looking to the fracture it i snecessary clarify on a statistical base the fracture differences. In fact, Probably for images quality, I cannot separate different phenomena and behaviour. 

When you speak about creep I am thinking to "time to fracture"..please add this parameter in order to correlate residual-stress to the creep resistance.

Moreover, the if the residual stresses have been  measured at room temperature, what is the temperature effect on the  different states? This point is really not clear to me.

I strongly believe that your is a valuable work, but it is necessary to improve it before the approval. 

Author Response

Dear reviewer:

I am very grateful to your comments for the manuscript. We also highly appreciate your carefulness, conscientious, and the broad knowledge on the relevant research fields, since you have given us a number of beneficial suggestions. According to the comments we received, we have made the revisions on this manuscript. 

The details of revisions have been upload.

Thank you very much

Dezheng Liu

Reviewer 2 Report

The authors investigated investigated the effect of prior residual stress on the creep rupture mechanism for P92 steel. The different prior residual stress levels were introduced into P92 high Cr alloy specimens using the local out-of-plane compression method. Subsequently the specimens were subjected to thermal exposure at the temperature of 650 °C for accelerated creep tests.

The creep rupture process under the different prior residual stress levels was analyzed by experimental observations and compared to FEM simulations. According to the authors transgranular cracking, resulting in ductile failure and plasticity-controlled cavity growth is the predominant mechanism when the introduced prior residual stress level is higher than the material intrinsic flow stress level. In contrast when the introduced prior residual stress level is lower than the material intrinsic flow stress level, intergranular dimple mixed cracking, resulting in brittle failure and nucleation-controlled constrained cavity growth is identified as the predominant mechanism.

The subject matter is suitable for publication in Metals but with respect to content and figures there are some small remarks for improvement:

Lines 46-48: There is a mistake in referencing within the sentence. Who is “most of them”?

Line 50-52: Sentence is unclear.

Figures 6 and 7: There is no comparability between figures 6 and 7 because of the different strain levels. Would it be possible to set the same strain levels for 6a and 7a, 6b and 7b and so on? It seems that there will be a long range strain influence for the lower stress level but probably it looks like only due to the different levels?

Figure 8: It would be better to use the same scale for the x-axes.

Figures 9 and 10: It would be better to combine both figures to only one. It is difficult to compare the microstructures if you have to switch from one page to the other.

References to related work are given.

The manuscript exhibits some linguistic and grammatical mistakes and needs revision.

Author Response

(The authors gave the same response as above.)

Reviewer 3 Report

The paper presents numerical and experimental methods to assess the influence of residual stresses on creep rupture mechanism for P92 steel. The subject is interesting both scientifically and industrially. Even if the paper is correctly written, some elements must be improved prior to publication:

- equation (1) and (2) are impossible to read on the pdf, maybe due to encoding issue, it must be corrected

- the value of the parameters used to derive curve presented in figure 2 must be given in the paper and a source must be cited. In addition, a deeper comment must be made on \beta coefficient ('an empiricial coefficient', what does it take into account and how is it determinde for the paper ?)

-page 6, line 181: it is stated that the parameters must be obtained from uniaxial creep behavior, but a reference work must be cited (if the values have been taken from the litterature) or the idenification method must be presented (if these have been obtained for the paper) --> see table 3

- equations (5) to (7) define quantities that doesn't seems to be used in the paper, are there typo in equation (4) (for example sij is used but Sij is defined later)

- page 6, line 184: \nu is defined but unused in the paper

- a flowchart may be usefull to help understanding the link between the diufferents softwares used in the paper; the link between Abaqus and hyperworks is unclear

- The type and the number of element in the FE model must be given and a comment must be done on the sesitivity of the results to mesh density, a comment must also be made on the time step

- figure 8 shows simulation results, what is the meaning of the error bars on the graphs ?

- a better comparison between experimental and numerical approach must b made in the conclusion

Concerning the language, several phrases are unclear and sometimes words are missing for a good comprehension for the reader. For example:

- page 1 line 32 'under the increasing pressures of power, economics and sustainability' what do you mean by 'power'

- page 2, line 49: '... its bottleneck was lack of accurate rupture mechanism...' is it 'was due to lack of ...' ?

- page 2 line 54: 'Due to the residual stress is inevitable...' is it 'Due to the fact that residual stress...' ?

and many other in the paper, please carrefully proofread the document.

Author Response

(The authors gave the same response as above.)

Round 2

Reviewer 1 Report

line 79: A non quoted reference still remains, probably a typing mistake… , please check it. 

Author Response

Dear reviewer:

I am very grateful for your comments and appreciate the careful and conscientious review you undertook on our behalf. I also appreciate your broad knowledge of the relevant research fields. Thank you very much for pointing out the typing mistake for us. According to your comments, the following revisions have been made to the manuscript:

Point 1: line 79: A non quoted reference still remains, probably a typing mistake… , please check it.

Response 1: Thank you for your carefully suggestion. Line 79: the typing mistake of the reference has been revised. Details see: line 79 in revised version. Furthermore, we have also carefully checked the sequence of the references number and the format of the paper.

Thank you very much for your comments on our paper (Manuscript ID. 515429). We feel lucky that our manuscript went to you as the valuable comments from you not only helped us with the improvement of our manuscript, but suggested some neat ideas for future studies. We are grateful for the opportunity to revise the manuscript and hope the revised version is acceptable for publication in Metals. Thank you very much for your review of the paper; if you have additional comments or questions, please do not hesitate to contact us.

Sincerely yours,

Dezheng Liu

Reviewer 3 Report

An intensive revision has been done, now the paper is suitable for publication. Thank you for the work.

Author Response

Dear reviewer:

I am very grateful for your comments and appreciate the careful and conscientious review you undertook on our behalf. I also appreciate your broad knowledge of the relevant research fields. Thank you very much for your review of the paper. According to your comments, the following responses have been made to the manuscript:

Point 1: An intensive revision has been done, now the paper is suitable for publication. Thank you for the work.

Response 1:

Thank you very much for your comments on our paper (Manuscript ID. 515429). We feel lucky that our manuscript went to you as the valuable comments from you not only helped us with the improvement of our manuscript, but suggested some neat ideas for future studies. We are grateful for the opportunity to revise the manuscript and hope the revised version is acceptable for publication in Metals. Thank you very much for your review of the paper; if you have additional comments or questions, please do not hesitate to contact us.

Sincerely yours,

Dezheng Liu
